# Extraordinary Characteristics of One-Dimensional PT-Symmetric Ring Optical Waveguide Networks Composed of Adjustable Length Ratio Waveguides

**DOI:** 10.3390/nano12193492

**Published:** 2022-10-06

**Authors:** Xian Liang, Xiangbo Yang, Jihui Ma, Mengli Huang, Dongmei Deng, Hongzhan Liu, Zhongchao Wei

**Affiliations:** Guangdong Provincial Key Laboratory of Nanophotonic Functional Materials and Devices, School of Information and Optoelectronic Science and Engineering, South China Normal University, Guangzhou 510006, China

**Keywords:** optical waveguide network, PT-symmetry, transmission, reflection, photonic location

## Abstract

A novel one-dimensional parity-time-symmetric periodic ring optical waveguide network (1D PTSPROWN) is constructed using magnesium fluoride (MgF_2_), by adjusting the length ratio of gain and loss materials in PT-symmetric waveguide and ordinary dielectric material, and by optimizing the program to search for the extremum spontaneous PT-symmetric breaking points. The ultra-strong transmission, reflection, and photonic location are noticed in the proposed 1DPTSPROWN as compared with the other PT-symmetric optical waveguide networks. The maximum and minimum reached 10^18^ and 10^−15^, respectively, which is more than 6 orders of magnitude greater and 3 orders of magnitude smaller than the best results reported so far. The ultra-strong transmission and reflection peaks, ultra-weak transmission, and reflection valleys generated by electromagnetic waves in this network were found to have interesting resonance and anti-resonance effects. Furthermore, frequency of periodic cycles and violet or redshift laws were discovered in the 1D PTSPROWN of fixed length ratio of gain and loss material in the PT-symmetric waveguide by adjusting the ratio of the upper and lower arm lengths of waveguides. The proposed optical waveguide network might have potential application in the design of CPA lasers, high-efficiency optical accumulators, and several other devices.

## 1. Introduction

In 1998, Bender et al. [1,2] proposed the concept of parity-time-symmetry (PT-symmetry) and found that when the Hamiltonian operator of a mechanical system of PT-symmetry is non-Hermitian and the imaginary part of its potential function is less than a certain critical value, all of its energy eigenvalues are complex. However, when the imaginary part of its potential function is greater than or equal to this critical value, its energy eigenvalue begins to appear as a real number. This critical value is called the spontaneous breaking point of the PT-symmetric system.

In 2007, El-Ganainy et al. [3] expanded the concept of PT-symmetric mechanical systems to PT-symmetric optical systems using the concept proposed in References [1,2] and constructing PT-symmetrical optical systems using gain and loss materials with a refractive index to meet $n(x)=n^{*}(-x)$ [4] requirements simultaneously. The potential function in quantum mechanics corresponds to the refractive index in the optical system under the condition of axial approximation, and the spontaneous breaking point of the PT-symmetric system corresponds to the imaginary part of the refractive index of the material of the PT-symmetric optical system. The PT-symmetric optical systems have several extraordinary optical properties, such as birefringence [4], super-strong transmission and reflection [5,6,7], super-strong photonic location [5,8,9,10], unidirectional and bidirectional invisibility [11,12,13,14,15,16,17], left-right misalignment [4,5,18], coherent perfect absorption [19,20,21,22,23,24], and so on. Thus, PT-symmetric optical systems have attracted widespread attention [3,4,5,6,7,8,9,10,11,12,13,14,15,16,17,18,19,20,21,22,23,24,25,26,27,28,29,30,31,32,33,34,35,36,37,38,39,40].

After photonic crystals, optical waveguide networks are considered another new type of artificial photonic bandgap micro-nano structure that can control and manipulate the phase and amplitude of electromagnetic waves, whose structures are more flexible and workable [41,42,43]. Recently, our group conducted a study of PT-symmetric optical waveguide networks and found that the ultra-strong extraordinary transmission, reflection, and photonic localization of the PT-symmetric photonic crystals were of the order of $10^{12}$ [5], which is 7 orders of magnitude greater than that of the photonic crystals reported so far. Since the PT-symmetric optical waveguide network can produce a much greater transmission, reflection, and photonic localization than the photon crystal, the question is whether the PT-symmetric optical waveguide network structure is further optimized to produce a greater ultra-strong transmission, reflection, and photonic location? Additionally, what interesting new properties will emerge in an optimized PT-symmetric optical waveguide network?

To address the aforementioned two issues, we have made two significant changes in two areas of this study, which are based on our group’s recent work [5]. First, the simulation program is optimized by reducing the calculation step length and improving the calculation accuracy, and the extremum spontaneous PT-symmetric breaking points of 1D PTSPROWN and the intrinsic frequencies of the system are more precisely determined. The singular optical properties of the optical waveguide network are generated near the extremum spontaneous PT-symmetric breaking points, and photon state mutations occur near them. As a result, even minor changes have a significant impact on the system’s resonance and anti-resonance effects, so we reduce the calculation step length and precisely determine the extremum spontaneous PT-symmetric breaking points, ensuring that the system’s resonance and anti-resonance effects are as strong as possible. Second, the unit-cell structural design is optimized by adjusting the length ratio of the upper and lower arm waveguides in the ring protocells and the length ratio of the ordinary dielectric materials and PT-symmetrical gain and loss materials in each waveguide. The optimal length ratios are then explored, and the extremum spontaneous PT-symmetric points of the corresponding systems are accurately determined. The purpose of regulating the length ratio of three waveguides in the unit-cell is to precisely regulate the natural frequency of the system so that it is closer to the frequency of the photon state mutation determined by the extremum spontaneous PT-symmetric breaking points. When they are more accurately matched, the electromagnetic wave resonance effect in the optical waveguide network system can be maximized.

It is discovered that by improving these two ways, the optimized system can produce super transmission, reflection, and photonic locations that are much larger than the original system, and the maximum value of them reached $10^{18}$, which is 6 orders of magnitude greater than the original maximum and the best result reported so far. The ultra-strong transmission and reflection peaks, ultra-weak transmission, and reflection valleys generated by electromagnetic waves in this network are found to have interesting resonance and anti-resonance effects. Furthermore, periodic coincidence laws, the violet (red) shift within the same period, and the violet (red) shift in different cycles were discovered in the 1D PTSPROWN of fixed length ratio of gain material and loss material in PT-symmetric waveguide by adjusting the ratio of the upper and lower arm lengths of waveguides. They are created by electromagnetic waves producing super-strong resonances and anti-resonances in PT-symmetric optical waveguide networks. These interesting properties might be used to design efficient photonic energy storage, extreme narrow-band optical filters, optical amplifiers, optical logic elements in photonic computers, and ultra-high monochromatic ultrasensitive optical switches.

This paper is organized as follows. Section 2 introduces the 1D PTSPROWN design model, and the used theoretical and numerical calculation methods. Section 3 defines the photonic pattern, determines its pattern distribution, draws the five extremum spontaneous PT-symmetric breaking points in the specified band, and thoroughly investigates the singular optical properties of the network. Finally, Section 4 gives a summary of this paper.

## 2. Materials and Theory

### 2.1. One-Dimensional PTSPROWN Modeling

The 1D PTSPROWN studied in this paper is based on the recently designed ring optical waveguide network of our group [5]. Its schematic diagram is shown in Figure 1.

To compare with the experimental results and develop practical applications, the working wavelength $\lambda_{\mathrm{work}}=c/\nu_{1}=d/7.297$ corresponding to the first extremum spontaneous PT-symmetric breaking point discussed in Section 2.6 is set by adjusting the waveguide length near the communication wavelength $\lambda_{\mathrm{com}}=1.550\;\mu m$, so $d_{2}=7.297\times \lambda_{\mathrm{com}}\approx 11.310\;\mu m$.

Comparing the work reported in Reference [5] and Figure 1, the resonance and anti-resonance effects of electromagnetic waves propagating in the optical waveguide network can be optimized by adjusting the ratio of the upper and lower arm lengths of waveguides in the ring unit-cell from the originally fixed ratio $d_{1}:d_{2}=1:1$ to
(1)$$d_{1}:d_{2}=1:G\;$$
where *G* is a positive integer. *G* can be a positive integers, positive fractions, and positive irrational numbers in practice. The focus of this paper was to make the system produce ultra-strong transmission and reflection. When *G* is a positive fraction or positive irrational number, the transmission and reflection properties of the system will be weakened. So, we chose *G* to be a positive integer. In addition, the ratio of the length of ordinary dielectric materials and PT-symmetrical gain and loss materials in each waveguide was changed from the originally fixed ratio $l_{1}:l_{2}:l_{3}=1:1:1$ to
(2)$$l_{1}:l_{2}:l_{3}=1:\eta :1\;$$
where η is a positive integer. This work mainly studies the extraordinary optical properties of 8 networks η=1−8. The optimal length ratio was decided by accurately identifying the disadvantages of the corresponding system’s extremum spontaneous PT-symmetric breaking point.

In Figure 1, the thick blue solid line in the unit-cell is a PT-symmetric waveguide consisting of MgF_2_. Its complex conjugate refractive index along the length of the waveguide can be achieved in two ways: by doping loss and gain of quantum dots or by changing the density of the material. The refractive indexes of the three sub-waveguides are:(3)$$\left\{\begin{array}{l} n_{1}=n_{{\mathrm{MgF}}_{2}}-in_{b}\\ n_{2}=n_{{\mathrm{MgF}}_{2}}\\ n_{3}=n_{{\mathrm{MgF}}_{2}}+in_{b} \end{array}\right.\;$$

For MgF_2_, it has been reported that the dispersion equation at 43–1500 THz satisfies the following equation [44]:(4)$$n_{{\mathrm{MgF}}_{2}}^{2}-1=\frac{0.48755108\lambda^{2}}{\lambda^{2}-0.04338408^{2}}+\frac{0.39875031\lambda^{2}}{\lambda^{2}-0.09461442^{2}}+\frac{2.312035\lambda^{2}}{\lambda^{2}-23.793604^{2}}\;$$

From Figure 2, we can see that the dispersion effect is weak in the range of 100–600 THz (0.5–3.0 μm). In order to reduce the influence of dispersion effect on the calculation results, we chose to study EM waves with the frequency (wavelength) in the range of 100–600 THz (0.5–3.0 μm).

### 2.2. Generalized Floquet–Bloch Theorem

It is well known that the Floquet–Bloch theorem is used for investigating lattice waves in a system with spatial translation periodicity. However, the waveguides of the optical waveguide network studied in this paper can be bent and folded arbitrarily, and the network does not necessarily have spatial translation periodicity, but there must be topological translation periodicity. Therefore, the dimensionless generalized Bloch theorem [45] was proposed to study optical waveguide networks:(5)$$\psi_{K}(\vec{N}+\vec{T})=\psi_{K}(\vec{N})e^{i\vec{K}\cdot \vec{T}}$$
where $\vec{K}$ is the structure Bloch wave vector, $\vec{N}$ is the node scale vector, $\vec{T}$ and is the structure translation vector; they are all dimensionless quantities. The generalized Bloch theorem was employed in this work to derive the dispersion relationship of the network, define the photon pattern, and determine the pattern distribution of photons.

### 2.3. Generalized Eigenfunction Method

The generalized eigenfunction method [46] was used to calculate the transmission, reflection, and photonic localization intensity of electromagnetic waves, which converts the wave transmission equations into a transmission matrix and treats the transmission coefficient and reflection coefficient as a generalized wave function during numerical calculation.

### 2.4. Three-Material Network Equation and Dispersion Relation

It is well-known that in a 1D PTSPROWN, the wave function between nodes can be considered as a linear combination of two opposite-direction plane waves given as:(6)$$\psi_{ij}\left(x\right)=\left\{\begin{array}{l} \psi_{n1}\left(x\right)=a_{1}e^{\iota k_{1}x}+b_{1}e^{-\iota k_{1}x}\;\left(0\leq x\leq l_{1}d_{ij}\right)\\ \psi_{n2}\left(x\right)=a_{2}e^{\iota k_{2}x}+b_{2}e^{-\iota k_{2}x}\left(l_{1}d_{ij}\leq x\leq l_{12}d_{ij}\right)\\ \psi_{n3}\left(x\right)=a_{3}e^{\iota k_{3}x}+b_{3}e^{-\iota k_{3}x}\left(l_{12}d_{ij}\leq x\leq l_{13}d_{ij}\right) \end{array}\right.\;$$
where the wave vector $k_{m}=2\pi \nu n_{m}/c(m=1,2,3)$, ν is the frequency, $l_{12}=l_{1}+l_{2}$, $n_{m}$ is the refractive index of the first sub-waveguide segment, c is the speed of light in the vacuum.

Through the conservation of flux energy, the 1D PTSPROWN equation [5] gives:(7)$$\sum_{j}\psi_{j}\frac{4k_{2}k_{3}}{\sum_{\alpha ,\beta }\Omega_{\alpha }\Xi_{\beta }{\left(-1\right)}^{\alpha +\beta }\sin d_{ij}{Ζ}_{\alpha ,\beta }}=\psi_{i}\sum_{j}\frac{\sum_{\alpha ,\beta }\Omega_{\alpha }\Xi_{\beta }{\left(-1\right)}^{\alpha +\beta }\cos d_{ij}{Ζ}_{\alpha ,\beta }}{\sum_{\alpha ,\beta }\Omega_{\alpha }\Xi_{\beta }{\left(-1\right)}^{\alpha +\beta }\sin d_{ij}{Ζ}_{\alpha ,\beta }}\;$$

Based on the generalized Bloch theorem and the three-material network equation, the dispersion relationship of the 1D PTSPROWN is given as:(8)$$f(k)=f(\frac{\omega }{c})=\cos K=\frac{\sum_{i}\frac{\sum_{\alpha ,\beta }\Omega_{\alpha }\Xi_{\beta }{\left(-1\right)}^{\alpha +\beta }\cos d_{ij}{Ζ}_{\alpha ,\beta }}{\sum_{\alpha ,\beta }\Omega_{\alpha }\Xi_{\beta }{\left(-1\right)}^{\alpha +\beta }\sin d_{ij}{Ζ}_{\alpha ,\beta }}}{\sum_{j=0}^{n}\frac{4k_{2}k_{3}}{\sum_{\alpha ,\beta }\Omega_{\alpha }\Xi_{\beta }{\left(-1\right)}^{\alpha +\beta }\sin d_{ij}{Ζ}_{\alpha ,\beta }}}$$
where
(9)$$\left\{\begin{array}{l} \alpha ,\beta =1,2\\ \Omega_{1}=k_{1}+k_{2}\\ \Omega_{2}=k_{1}-k_{2}\\ \Xi_{1}=k_{3}+k_{2}\\ \Xi_{2}=k_{3}-k_{2}\\ {Ζ}_{1,1}=k_{1}l_{1}+k_{2}l_{2}+k_{3}l_{3}\\ {Ζ}_{2,1}=k_{1}l_{1}-k_{2}l_{2}-k_{3}l_{3}\\ {Ζ}_{1,2}=k_{1}l_{1}+k_{2}l_{2}-k_{3}l_{3}\\ {Ζ}_{2,2}=k_{1}l_{1}-k_{2}l_{2}+k_{3}l_{3} \end{array}\right.$$

### 2.5. Extraordinary Photonic Modes

Since the waveguides of the optical waveguide network studied in this paper can be bent and folded arbitrarily, the network does not necessarily have spatial translation periodicity, but there must be topological translation periodicity. Therefore, the optical waveguide networks follow the dimensionless generalized Bloch theorem: $\psi_{K}(\vec{N}+\vec{T})=\psi_{K}(\vec{N})e^{i\vec{K}\cdot \vec{T}}$. (i) APM: When $K=K_{a}+iK_{b}\;(K_{b}>0)$ is a complex number, $i\vec{K}\cdot \vec{T}=iK_{a}T-K_{b}T$. When EM waves propagate in this kind of network, not only the phase of the wave function changes with a factor, $e^{iK_{a}T}$, but also the amplitude of the wave function attenuates with a factor, $e^{-iK_{b}T}$. We call this kind of photonic mode attenuation propagation mode (APM). (ii) GPM: When $K=K_{a}-iK_{b}\;(K_{b}>0)$ is a complex number, $i\vec{K}\cdot \vec{T}=iK_{a}T+K_{b}T$. When EM waves propagate in this kind of network, not only the phase of the wave function changes with a factor, $e^{iK_{a}T}$, but also the amplitude of the wave function gains with a factor, $e^{iK_{b}}{}^{T}$. We call this kind of photonic mode gain propagation mode (GPM). (iii) When f(ν)>1, both APM and GPM produced by the PT-symmetric waveguide network are called strong propagation modes (SPMs).

For the 1D PTSPROWN studied in this paper, the dispersion function f(v) is complex in the full frequency band because the refractive index of two materials is complex. Mathematically, when f(v) is plural, there is no real solution of the Bloch wave vector $\vec{K}$ but a pair of conjugated complex solutions. Physically, the 1D PTSPROWN studied in this paper has both a gain mechanism and an attenuation mechanism. The negative imaginary solution of the dispersion function f(v) corresponds to the gain propagation mode and the positive imaginary solution of the dispersion function f(v) corresponds to the attenuation propagation mode. Traditional APM and GPM do not exist in PT-symmetric optical waveguide networks, and traditional band structure diagrams are no longer applicable, so we need to define new photonic patterns and draw corresponding photonic pattern distribution maps.

For 1D PTSPROWN, the dispersion relation (8) shows: (i) when |f(v)|≤1, the virtual part of the structure Bloch wave vector $\vec{K}$ is small and cannot produce an extraordinary transmission/reflection greater than 1.0; (ii) when |f(v)|>1, the virtual part of the structure Bloch wave vector $\vec{K}$ is large and can produce an extraordinary transmission/reflection greater than 1.0. We used |f(v)|=1 as the photonic mode demarcation point for a PT-symmetrical optical waveguide network. The APM and GPM generated by the waveguide network are determined by the corresponding structure Bloch wave vectors and are defined as weak modes and referred to as WPM [5]. Whereas the APM and GPM generated by the waveguide network are determined by the corresponding structure Bloch wave vectors and are defined as strong propagation mode and are referred to as SPM [5]. The extremum spontaneous PT-symmetric breaking point of the 1D PTSPROWN studied in this paper was defined as the value of the imaginary part of the refractive index of the material at the junction of WPM and SPM. A 1D PTSPROWN determined by the extremum spontaneous PT-symmetric breaking point produces extraordinary optical properties. This paper investigated the singular optical properties of electromagnetic waves in the frequency range of 100–600 THz considering the dispersion effect of MgF_2_ and the disadvantages of extreme spontaneous PT-symmetric breaking of extreme values in the 1D PTSPROWN and the extreme spontaneous PT-symmetric breaking disadvantages of 1D PTSPROWN defined in Section 2.5. The plot of a 1D PTSPROWN photonic pattern distribution of G=1 is shown in Figure 3, where the red and white areas represent SPMs and WPMs, respectively.

### 2.6. Extremum Spontaneous PT-Symmetric Breaking Points

From Figure 3, it can be seen that for 1D PTSPROWN of G=1, there are 5 extremum spontaneous PT-symmetric breaking points in the 100–600 THz frequency band, and the frequency and the values of the imaginary part of the refractive corresponding to these 5 extremum spontaneous breaking points are as follows:(10)$$\left\{\begin{array}{l} v_{I}=193.414999986\mathrm{THz},\;n_{bI}=2.280\times 10^{-9}\\ v_{\mathrm{II}}=272.825872840\mathrm{THz},\;n_{\mathrm{bII}}=2.758\times 10^{-7}\\ v_{\mathrm{III}}=385.497799758\mathrm{THz},\;n_{\mathrm{bIII}}=2.210\times 10^{-9}\\ v_{\mathrm{IV}}=481.215799958\mathrm{THz},\;n_{\mathrm{bIV}}=2.385\times 10^{-9}\\ v_{V}=576.583199823\mathrm{THz},\;n_{\mathrm{bV}}=4.652\times 10^{-7} \end{array}\right.$$

It should be noted that before determining the specific length *d*, we set *d* equal to the unit length, and the frequencies corresponding to the 5 extremum spontaneous PT-symmetric points of 1D PTSPROWN are:(11)$$\left\{\begin{array}{l} \nu_{1}=7.297c/d\\ \nu_{2}=10.29c/d\\ \nu_{3}=15.54c/d\\ \nu_{4}=18.15c/d\\ \nu_{5}=21.75c/d \end{array}\right.\;$$

If the frequency of the first extremum spontaneous PT-symmetric breaking point is taken as the operating frequency, then its corresponding operating wavelength is near the communication wavelength. The specific wavelength *d* is given as d=11.31 μm, according to the model described in the first paragraph of Section 2.1.

We chose the frequency corresponding to the first extremum spontaneous PT-symmetric breaking point as the operating frequency because it can be calculated that the maximum transmission and maximum photonic localization strength generated by the network are greater than the network determined by the other 4 extremum spontaneous PT-symmetric breaking points by using the first extremum spontaneous PT-symmetric breaking defect as the virtual part of the refractive index of the waveguide material as shown in Figure 1. Therefore, the first extremum spontaneous PT-symmetry breaking point was used to determine the virtual part of the refractive index of the gain and loss materials to improve the performance of the 1D PTSPROWN.

## 3. Results

In this section, all results were under ideal conditions, and next we will discuss an evaluation of the impact of expected non-idealities in the following two aspects.

Imprecision in refractive index values: Through calculations, we found that with the change of light intensity, considering the influence of nonlinear effects, if the accuracy of the refractive index only changes by less than or equal to 10^−2^ orders of magnitude, the system has good robustness. Thus, the nonlinear effect of light intensity will not have an impact on the transmission and photon local order of magnitude. Taking the published papers of our group [29] as an example, the refractive index of BBO is $n_{\mathrm{BBO}}=1.66+n_{2}I,n_{2}=\mp 8\times 10^{-}{}^{20}m^{2}/V$, where $n_{2}$ is the nonlinear coefficient and *I* is the intensity of the electromagnetic wave. The incident electromagnetic wave intensity *I* is small, and after multiplying a nonlinear coefficient $n_{2}$ of order of magnitude 10^−20^, the resulting real number $n_{2}I$ is still a small value far less than 10^−2^. In this paper, the maximum photonic localization is 10^18^, and after multiplying the nonlinear coefficient $n_{2}$, we still obtained a value of less than 10^−2^. Therefore, we did not take the nonlinear effects of the real network into consideration in this paper, and our results and measured values will not be different too much. In practice, if a material with a smaller nonlinear effect has less effect on the refractive index value, better results will be obtained.

Variation in the lengths of each section: In this paper, the ratio of the upper and lower arm lengths of waveguides was 1:*G*, where *G* can be both integer and non-integer. In practice, if *G* changes from an integer to a non-integer, the position and shape of the photonic bands produced in the optical waveguide network change slightly, but the arrangement order of the passband and bandgap remains the same, and the transmission and reflection changes slightly, but the order of magnitude remains the same. That is, if *G* changes from an even number to a non-integer number, a wide bandgap will become one extremely narrow pass band and two extremely narrow bands, and if *G* changes from an odd number to a non-integer number, a wide pass band will become one extremely narrow band and two narrower pass bands. Whether it is an odd-number-broken or an even-number-broken, the system will only produce photonic band position and shape changes, without affecting the order of magnitude of transmission, reflection, and photonic localization. Taking the published papers of our group [46,47] as an example, when the ratio of the upper and lower arm lengths of waveguides is *d*_2_:*d*_1_ = 6:1, 5.8:1, and 6.2:1, respectively, the position and shape of the photon bands produced in the optical waveguide network change slightly, but the transmission is almost unchanged. Therefore, we did not take the case that *G* is a non-integer into consideration in this paper, and our results and measured values will not be different too much.

In summary, both imprecision in refractive index values and variation in the lengths of each section will have an effect on the transmittance and photon locality generated by the system, but not on orders of magnitude, so the whole system has good robustness.

### 3.1. One-Dimensional PTSPROWN of G=1

#### 3.1.1. Extraordinary Transmission and Reflection

As compared to the original model [5], the ultra-strong reflection of the new model has been greatly enhanced after the optimization of η in the model structure given in Section 2.1. After calculating the reflection at the five extremum spontaneous PT-symmetric breaking points in the photonic mode distribution plot at different cell numbers, we found that in the frequency band of 100–600 THz, the 1DPTSPROWN determined by the first extremum spontaneous PT-symmetric breaking point produced the strongest transmission (reflection).

In this section, we studied the reflecting properties of the 1D PTSPROWN of G=1, as determined by the first extremum spontaneous PT-symmetric breaking point by varying the parameter η in Equation (2). We found that the translative and reflective properties of the optical waveguide network computed η as an integer were significantly improved than those that determined η as a decimal. So, we only studied the optical waveguide network whose η was an integer. The maximum transmission (reflection) and the minimum transmission (reflection) of 1D PTSPROWN corresponding to the η=1−8 are shown in Table 1. As it can be seen:

(i) Ultra-strong coupled resonance and anti-resonance effects: The electromagnetic waves of frequencies near the extremum spontaneous breaking points can produce extraordinary ultra-strong transmission and reflection peaks as well as ultra-weak transmission and reflection valleys in optical waveguide networks whether it is a left or right incident. Taking system No. 5 in Table 1 as an example, when η=5, the left-incident transmission (reflection) peak reaches $1.462\times 10^{18}$, which is 6 orders of magnitude greater than the best-reported result, and the right-incident transmission (reflection) peak reaches $5.437\times 10^{13}$, which is 1 order of magnitude greater than the best-reported result [5], showing the ultra-strong coupling resonance effect. Furthermore, the ultra-weak transmission valley value reaches $2.276\times 10^{-15}$, which is 3 orders of magnitude smaller than the best result [5] reported so far, showing the ultra-strong coupled anti-resonance effect.

(ii) Different coupling resonance effects: The peaks of left (right) transmission and reflection are different in the network systems determined at various values of η. Taking system No. 1 and system No. 5 from Table 1 as examples, the left-incident transmission (reflection) peak reaches $3.715\times 10^{14}$ when η=1, and the left-incident transmission (reflection) peak reaches $1.462\times 10^{18}$ when η=5. Consequently, for systems determined by different η, the coupling resonance effect of systems determined for various η is different.

(iii) Left-right asymmetry: The peaks of left (right) transmission and reflection generated by left-incident and right-incident electromagnetic waves in optical waveguide networks do not coincide, showing the different locations and unequal peak values. Taking system No. 5 in Table 1 as an example, the left-incident transmission (reflection) peak reaches $1.462\times 10^{18}$ and the right-incident transmission (reflection) peak reaches $5.437\times 10^{13}$ when η=5. Consequently, the peak values are different and left-right incompatibility is evident.

#### 3.1.2. Ultra-Strong Photonic Localization

After the optimization of η in the model structure proposed in Section 2.1, the photonic localization of the new model has been greatly enhanced when compared with the original model [5].

The intensity map of 1D PTSPROWN whose G=1 and η=5, determined by the first extremum spontaneous PT-symmetric breaking point using the generalized eigenfunction method is shown in Figure 4. When G=1, the upper and lower wave conductors are symmetrical, and the intensity distribution of the upper and lower wave conductors is the same, so we only need to draw the intensity distribution of either the upper or the lower wave wire. As shown in Figure 4, *i*-*d*-*j* represents the upper/lower waveguide between node *i* and node *j*.

As can be seen in Figure 4, the electromagnetic wave intensity in the upper or lower arm of the unit-cell appears in the middle and both ends of the waveguide at the frequency of ν=193.41578571903 THz, and the maximum intensity of the ultra-strong photonic localization reaches $1.462\times 10^{18}$. The electromagnetic wave intensity distributions in the three protocells are very similar; they all contain three peaks and two valleys, and the placements and values of the peaks and valleys are identical. We hypothesized that the ultra-strong and the ultra-weak photonic localization were caused by the gain and loss structure in 1D PTSPROWN, which produces an ultra-strong coupling resonance effect. In terms of practical applications, the ultra-strong photonic localization properties of the 1D PTSPROWN may be used to create high-efficiency photonic energy accumulators and high-power light-emitting diodes.

### 3.2. One-Dimensional PTSPROWN of G=1,2, 3⋯

#### 3.2.1. Resonance and Anti-resonance Effects

When the photon pattern distribution map of 1D PTSPROWN was studied for G=1, 2,  3⋯, the photon pattern distribution map of G=1 only has the extremum spontaneous PT-symmetric breaking points, and the 1D PTSPROWNs for G(G=2, 3…) decided by the extremum spontaneous PT-symmetric breaking points also have extraordinary ultra-strong transmission and reflection properties. Consequently, in this section, we determined the 1D PTSPROWN system by the first extremum spontaneous PT-symmetric breaking points, and G=1.

After optimizing the η in model structure given in Section 2.1, the new system can produce not only the ultra-strong transmission, reflection, and photonic localization, but also produce ultra-strong coupled resonance and anti-resonance effects as compared to the original system [5]. All the 1D PTSPROWNs are determined by G and η and follow the following laws.

(i)Resonance effect: The transmission and reflection peaks of the left (right) incidence coincide perfectly in terms of the position and amplitude of the peaks. Consider for example the 1D PTSPROWN of G=1 and η=5. As can be seen in Figure 5, when the frequency ν=193.415785715 THz, $T_{L\max}=R_{L}{}_{max}=1.462\times 10^{18}$ and when the frequency ν=193.415786465 THz, $T_{L\max}=R_{L}{}_{max}=5.437\times 10^{13}$**.** The transmission (reflection) peak coincides perfectly, thus exhibiting an ultra-strong resonance effect.(ii)Anti-resonance effect: The transmission and reflection valleys of the left (right) incidence coincide perfectly in terms of position and amplitude. Consider for example the 1D PTSPROWN whose G=1 and η=5. As can be seen in Figure 5, when the frequency ν=193.415785715 THz, $T_{L\min}=R_{\mathrm{Lmin}}=2.276\times 10^{-15}$. The transmission (reflection) valley coincides perfectly, thus exhibiting an ultra-strong anti-resonance effect.

#### 3.2.2. Periodicity in the Transmission Spectra

In addition to the above-mentioned extraordinary optical properties, it was also found that after model optimization of η and G in the model structure given in Section 2.1, the position of the ultra-strong transmission and reflection peaks produced by 1D PTSPROWN with the same number of cells N have three additional interesting properties as the increment of G: the periodic coincidence law, the violet (red) shift law in the same period, and the violet (red) shift law in different periods.

(i) Periodic coincidence law: For a given value of η, as the increment of G, the position of the transmission (reflection) peak of the left (right) incident will coincide periodically, but the peaks are different, as shown in columns 1, 2, and 3 of Table 2. As an example, we introduced a rule in the first line of Table 2: when η=1 and G=3P(=1,2,…,), the transmission (reflection) peaks coincide as P increases.

(ii) Violet (red) shift law in the same period: For a given value of η, as the increment of G, the position of the left (right) incident transmission (reflection) peaks show the violet (red) shift phenomenon in the same period concerning columns 1, 2, 4, and 6 of Table 2. As an example, we introduced the following law in the first line of Table 2: when η=1, the transmission (reflection) peaks of the left (right) incident in the same period of P presents a purple (red) shift in the order of 3P-2, 3P-1, 3P.

(iii) Violet (red) shift law in different cycles: For a given value of η, as the increment of G, the position of the left (right) incident transmission (reflection) peaks show a purple (red) shift in different periods, regarding columns 1, 2, 5, and 7 of Table 2. We introduced the following rule in the first line of Table 2: when η=1, for different cycles of P that are not the same, the transmission (reflection) peaks of the left (right) present a purple (red) shift in the order of 3P-R, 3Q-R (P < Q).

#### 3.2.3. Verifying Periodic Coincidence Laws by Numerical Results

In Section 3.2.2, the frequency distribution of the transmission and reflection peaks (valleys) were studied in detail. The periodic coincidence law of the transmission and reflection peaks, as the increment of G in the 1D PTSPROWNs for the given η, were deduced. Numerically, taking the two 1D PTSPROWNs determined by η=1 and η=2 to draw a left (right) transmission spectrum that is shown in Figure 6, when η=1 and G=3(6), the 1D PTSPROWN left (right) transmission peaks coincide; when η=1 and G=2(4), the 1D PTSPROWN left (right) transmission and reflection peaks coincide, so the periodic coincidence law is verified by the numerical results.

#### 3.2.4. Verifying Periodic Coincidence Laws by Analytical Derivation

To numerically verify the cycle coincidence law, let us take the 1D PTWPROWN of η=1 and G=3P as an example. Given by Equation (8), we derive the equation:(12)$$\begin{array}{cc} f(k) & =\frac{\sum_{i}\frac{\sum_{\alpha ,\beta }\Omega_{\alpha }\Xi_{\beta }{\left(-1\right)}^{\alpha +\beta }\cos d_{ij}{Ζ}_{\alpha ,\beta }}{\sum_{\alpha ,\beta }\Omega_{\alpha }\Xi_{\beta }{\left(-1\right)}^{\alpha +\beta }\sin d_{ij}{Ζ}_{\alpha ,\beta }}}{\sum_{j=0}^{n}\frac{4k_{2}k_{3}}{\sum_{\alpha ,\beta }\Omega_{\alpha }\Xi_{\beta }{\left(-1\right)}^{\alpha +\beta }\sin d_{ij}{Ζ}_{\alpha ,\beta }}}\\ & =\frac{H_{3}/H_{1}+H_{4}/H_{2}}{4k_{2}k_{3}H_{1}+H_{2}}\\ & =\frac{H_{2}H_{3}+H_{4}H_{1}}{4k_{2}k_{3}\left(H_{1}+H_{2}\right)} \end{array}$$
where
(13)$$\left\{\begin{array}{l} H_{1}=A_{1}B_{1}\sin\left(d_{1}M_{1}\right)+A_{1}B_{2}\sin\left(d_{1}M_{2}\right)-A_{2}B_{1}\sin\left(d_{1}M_{3}\right)-A_{2}B_{2}\sin\left(d_{1}M_{4}\right)\\ H_{2}=A_{1}B_{1}\sin\left(d_{2}M_{5}\right)+A_{1}B_{2}\sin\left(d_{2}M_{6}\right)-A_{2}B_{1}\sin\left(d_{2}M_{7}\right)-A_{2}B_{2}\sin\left(d_{2}M_{8}\right)\\ H_{3}=A_{1}B_{1}\cos\left(d_{1}M_{1}\right)+A_{1}B_{2}\cos\left(d_{1}M_{2}\right)-A_{2}B_{1}\cos\left(d_{1}M_{3}\right)-A_{2}B_{2}\cos\left(d_{1}M_{4}\right)\\ H_{4}=A_{1}B_{1}\cos\left(d_{2}M_{5}\right)+A_{1}B_{2}\cos\left(d_{2}M_{6}\right)-A_{2}B_{1}\cos\left(d_{2}M_{7}\right)-A_{2}B_{2}\cos\left(d_{2}M_{8}\right) \end{array}\right.$$
(14)$$\left\{\begin{array}{l} k_{1}=2\pi n_{1}v/c\\ k_{2}=2\pi n_{2}v/c\\ k_{3}=2\pi n_{3}v/c\\ A_{1}=k_{1}+k_{2}\\ A_{2}=k_{1}-k_{2}\\ B_{1}=k_{3}+k_{2}\\ B_{2}=k3-k_{2} \end{array}\right.$$
(15)$$\left\{\begin{array}{l} M_{1}=k_{1}l_{11}+k_{2}l_{12}+k_{3}l_{13}\\ M_{2}=k_{1}l_{11}+k_{2}l_{12}-k_{3}l_{13}\\ M_{3}=k_{1}l_{11}-k_{2}l_{12}-k_{3}l_{13}\\ M_{4}=k_{1}l_{11}-k_{2}l_{12}+k_{3}l_{13}\\ M_{5}=k_{1}l_{21}+k_{2}l_{22}+k_{3}l_{23}\\ M_{6}=k_{1}l_{21}+k_{2}l_{22}-k_{3}l_{23}\\ M_{7}=k_{1}l_{21}-k_{2}l_{22}-k_{3}l_{23}\\ M_{8}=k_{1}l_{21}-k_{2}l_{22}+k_{3}l_{23} \end{array}\right.$$

From Equation (12), it can be seen that with the increment of P, only the phase in Equation (13) of 1D PTSPROWN for η=1 will be affected, so we directly calculated the change of phase ($d_{1}M_{1},d_{1}M_{2},d_{1}M_{3},d_{1}M_{4}$) in Equation (13), that is $\Delta \varphi_{1},\Delta \varphi_{2},\Delta \varphi_{3},\Delta \varphi_{4}$, and the calculation states:

(i)When G changes from 3 to 6 (P changes from 1 to 2), the corresponding changes of phase are $\Delta \varphi_{1}=54\pi ,\Delta \varphi_{2}=18\pi ,\Delta \varphi_{3}=-18\pi ,\Delta \varphi_{4}=18\pi$, respectively.(ii)When G changes from 3 to 9 (P changes from 1 to 3), the corresponding changes of phase are $\Delta \varphi_{1}=144\pi ,\Delta \varphi_{2}=48\pi ,\Delta \varphi_{3}=-48\pi ,\Delta \varphi_{4}=48\pi$, respectively.

In summary, when G changes from 3 to 3P, the phase changes of the 1D PTSPROWN for η=1 determined by the first extremum spontaneous breaking point are all integer multiples of 2π. The transmission (reflection) peak corresponding to frequencies are equal, and the left (right) transmission and reflection peak positions in the transmission spectra coincide. Consequently, the periodic coincidence law was verified. Similarly, the period coincidence law of the left (right) transmission peaks position of 1D PTSPROWN can be verified.

## 4. Conclusions

A novel one-dimensional parity-time-symmetric periodic ring optical waveguide network (1D PTSPROWN) was constructed using magnesium fluoride (MgF_2_), by adjusting the length ratio of gain material and loss material in the PT-symmetric waveguide and ordinary dielectric material and by optimizing the extremum spontaneous PT-symmetric breaking points. Two important improvements have been made based on the recent work of our group. First, the calculation technique was optimized by reducing the calculation step length and improving the calculation accuracy, and the extremum spontaneous PT-symmetric breaking points of the PT-symmetric optical waveguide network were accurately explored and a more accurate intrinsic frequency of the system was achieved. Second, the structure design of the unit-cell was optimized by adjusting the length proportion of the ordinary medium material and the gain and loss material of PT-symmetry in each waveguide. Furthermore, the extremum spontaneous PT-symmetric breaking points of the corresponding system were accurately determined.

Through the optimization of the model structure, the ultra-strong transmission and reflection of the new system have been significantly enhanced as compared to the original system. We discovered that the optimized system can produce far greater transmission, reflection, and photonic localization than the original system. The maximum value of transmissivity reached $1.462\times 10^{18}$, which is 6 orders of magnitude greater than the best result reported so far. These extraordinary results are credited to the accurate search of the extremum spontaneous PT-symmetric breaking points of the 1D PTSPROWN to obtain the precise intrinsic eigenfrequency of the 1D PTSPROWN by adjusting the proportion of the length of the ordinary dielectric material and the PT-symmetric gain and loss material in each waveguide. As a result, the electromagnetic waves have an ultra-strong resonance effect as they propagate through the system.

Furthermore, the ultra-strong transmission and reflection peaks, ultra-weak transmission, and reflection valleys generated by electromagnetic waves in this network were found to have interesting resonance and anti-resonance effects.

Additionally, the 1D PTSPROWN of fixed length ratio of gain material and loss material in the PT-symmetric waveguide by adjusting the ratio of the upper and lower arm lengths of waveguides was studied. It was observed that the ultra-strong transmission and reflection peaks, ultra-weak transmission, and reflection valleys produced by the optimized system with the change of upper and lower arm ratios of wave conductors have the following additional extraordinary optical properties: resonance and anti-resonance effects, the periodic coincidence law, the violet (red) shift law in the same period, and the violet (red) shift law in different periods. These extraordinary optical properties make it possible to select the best optical model at the specified wavelength. These intriguing properties might be used in the development of efficient photon energy storage, extreme narrowband optical filters, optical amplifiers, optical logic elements in photonic computers, and ultra-high monochromatic ultrasensitive optical switches.

## Figures and Tables

**Figure 1 nanomaterials-12-03492-f001:**
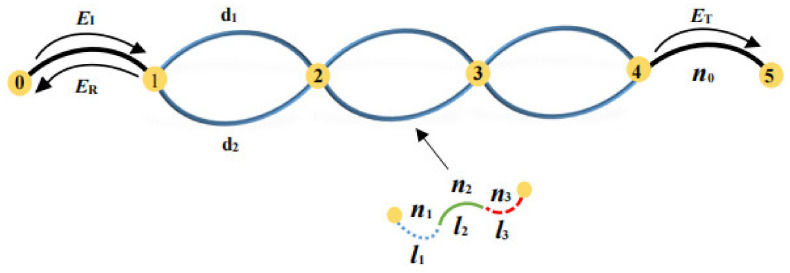
Schematic diagram of the 1D PTSPROWN with one entrance, one exit, and 3 unit-cells, where ***E_I_***, ***E_R_***, and ***E_T_*** are the input, reflective, and output electromagnetic waves, respectively. The thick black solid lines at the entrance and outlet are the vacuum optical waveguide, and their length is *d.* The thick blue solid lines in the unit-cell made up of three sub-waveguides is a PT-symmetric waveguide, and the length of the waveguides is *d*_1_ and *d*_2_. The refractive indices of the three sub-waveguides are, respectively, *n*_1_, *n*_2_, and *n*_3_, and their length ratios are *l*_1_, *l*_2_, and *l*_3_, respectively.

**Figure 2 nanomaterials-12-03492-f002:**
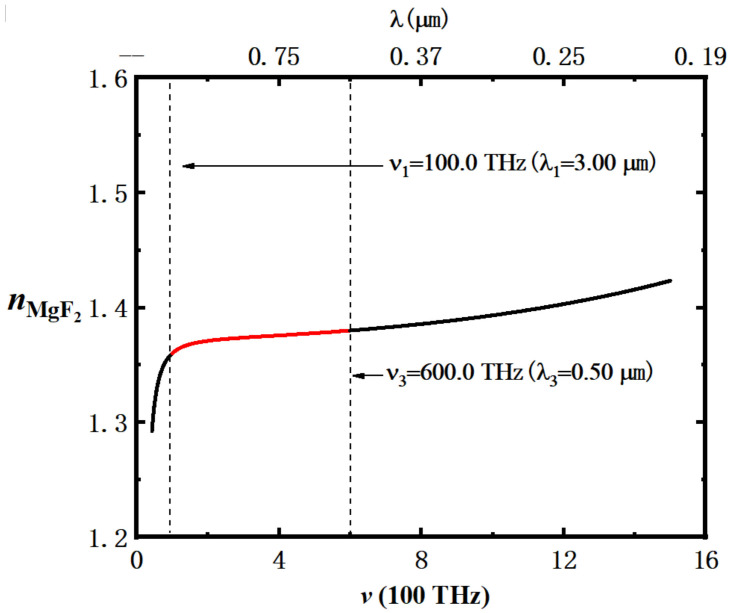
The dispersion curve of MgF_2_ defined by Equation (4): the red curve indicates the frequency range studied in this paper, and the black thin dashed line is the scale line at the frequency endpoint and communication wavelength studied in this article, and its values are $v_{1}=100.0\;\mathrm{THz}\;(\lambda_{2}=3.0\;\mu m)$, $v_{3}=600.0\;\mathrm{THz}\;(\lambda_{3}=0.5\;\mu m)$, respectively.

**Figure 3 nanomaterials-12-03492-f003:**
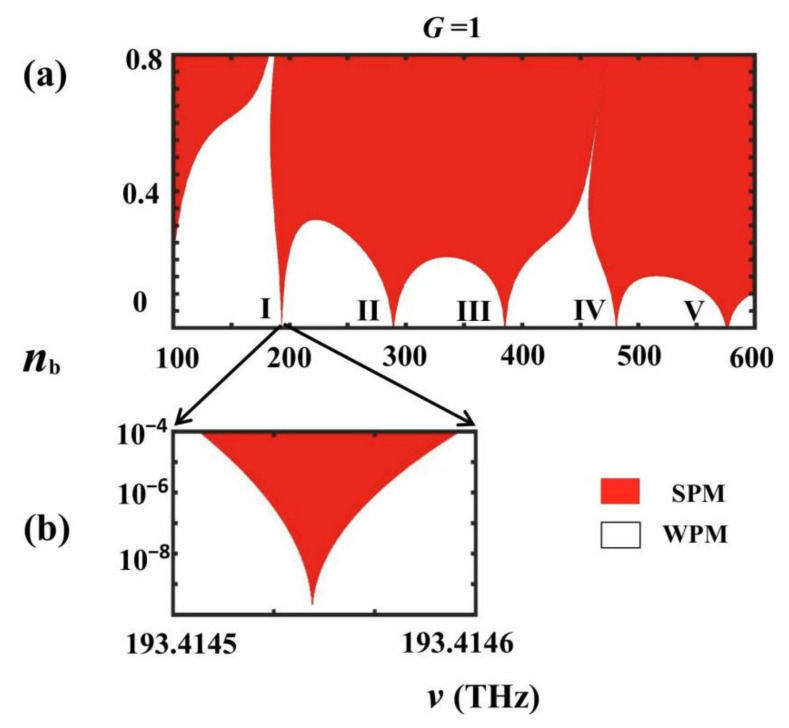
Photonic pattern distribution map of 1D PTSPOWN of G=1, where the white and red areas represent weak modes (WPMs) and strong modes (SPMs), respectively: (**a**) a general map of the frequency (wavelength) range corresponding to the thick red line in Figure 2; (**b**) an enlarged view of the first extremum spontaneous PT-symmetric breaking point near the communication wavelength.

**Figure 4 nanomaterials-12-03492-f004:**
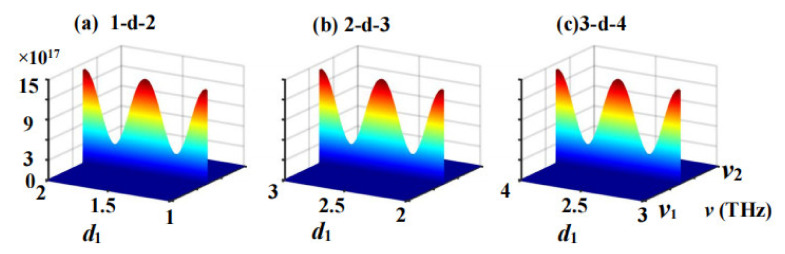
Intensity map of the photonic location of the 1D SPROWN of G=1 and η=5,determined by the first extremum spontaneous PT-symmetric breaking point, where *i*-*d*-*j* (*i*,*j* = 1,2,3) indicates the upper (lower) arm waveguide between nodes *i* and node *j*, $\nu_{1}=193.415785719\mathrm{THz}$, $\nu_{2}=193.41578571906\mathrm{THz}$: (**a**) the intensity distribution pattern in the 1-*d*-2 waveguide in Figure 1; (**b**) the intensity distribution pattern in the 2-*d*-3 waveguide in Figure 1; (**c**) the intensity distribution pattern in the 3-*d*-4 waveguide in Figure 1.

**Figure 5 nanomaterials-12-03492-f005:**
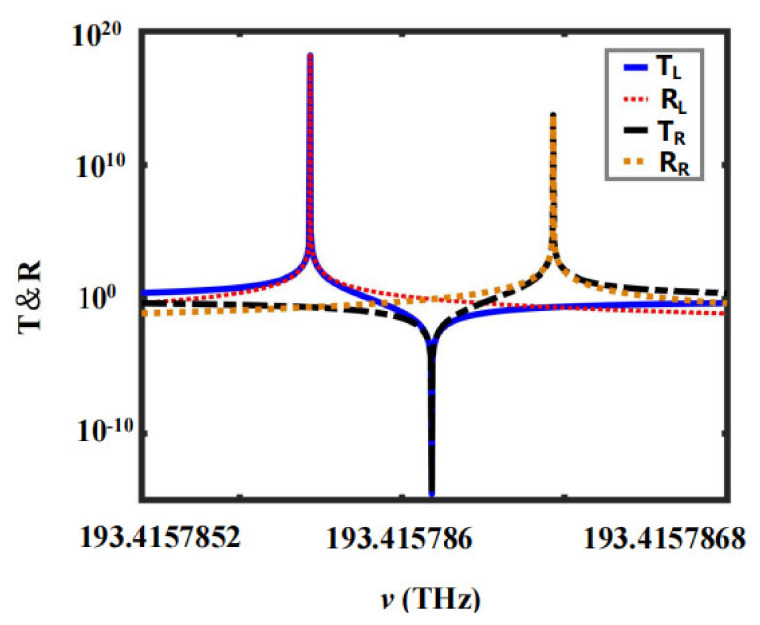
The transmission and reflection spectra of 1D PTSPROWN of G=1 and η=5, determined by the first extremum spontaneous PT-symmetric breaking point. The blue solid line and the red dotted line represent the left-incident transmission and left-incident reflection, respectively. The black point dotted line and the orange point line represent the right-incident transmission and right-incident reflection, respectively.

**Figure 6 nanomaterials-12-03492-f006:**
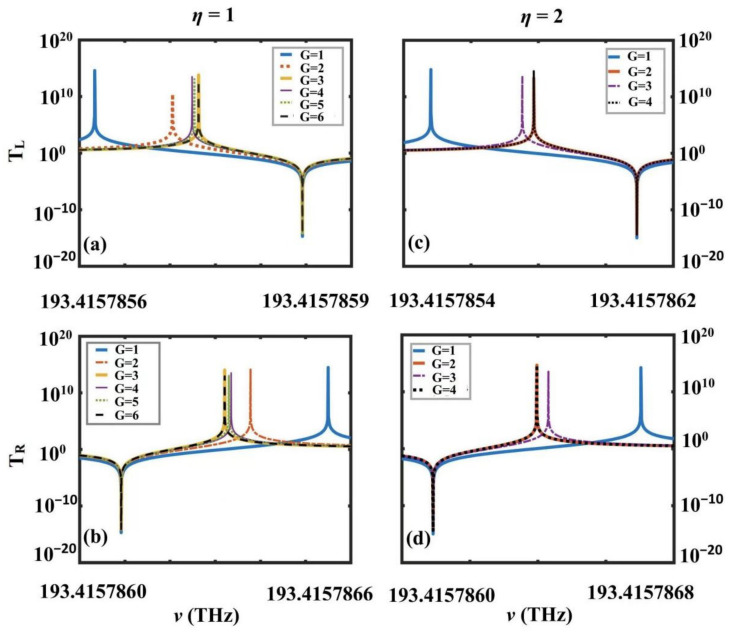
Transmission spectra of 1D PTSPROWN of η (η=1,2), decided by the first extremum spontaneous PT-symmetric breaking point in two change periods of G: (**a**,**b**) η=1, *G* = 1–6; (**c**,**d**), *G* = 1–4.

**Table 1 nanomaterials-12-03492-t001:** Left (right) transmission and reflection peak/valley value of 8 1D PTSPROWNs of G=1 and η=1−8, as determined by the first extremum spontaneous PT-symmetric breaking points.

**No**	** *η* **	*T_Lmax_*	*R_Lmax_*	*T_Lmin_*	*T_Rmax_*	*R_Rmax_*	*T_Rmin_*
1	1	$3.715\times 10^{14}$	$3.745\times 10^{14}$	$1.816\times 10^{-15}$	$1.669\times 10^{14}$	$1.669\times 10^{14}$	$1.816\times 10^{-15}$
2	2	$6.174\times 10^{14}$	$6.174\times 10^{14}$	$1.009\times 10^{-15}$	$1.554\times 10^{14}$	$1.554\times 10^{14}$	$1.009\times 10^{-15}$
3	3	$2.172\times 10^{15}$	$2.172\times 10^{15}$	$1.245\times 10^{-15}$	$8.520\times 10^{14}$	$8.520\times 10^{14}$	$1.245\times 10^{-15}$
4	4	$3.507\times 10^{15}$	$3.507\times 10^{15}$	$1.822\times 10^{-15}$	$2.539\times 10^{14}$	$2.539\times 10^{14}$	$1.822\times 10^{-15}$
5	5	$1.462\times 10^{18}$	$1.462\times 10^{18}$	$2.276\times 10^{-15}$	$5.437\times 10^{13}$	$5.437\times 10^{13}$	$2.276\times 10^{-15}$
6	6	$7.639\times 10^{16}$	$7.639\times 10^{16}$	$5.404\times 10^{-15}$	$4.422\times 10^{15}$	$4.422\times 10^{15}$	$5.404\times 10^{-15}$
7	7	$1.702\times 10^{17}$	$1.702\times 10^{17}$	$6.059\times 10^{-15}$	$2.345\times 10^{15}$	$2.345\times 10^{15}$	$6.059\times 10^{-15}$
8	8	$4.489\times 10^{15}$	$4.489\times 10^{15}$	$8.671\times 10^{-15}$	$5.915\times 10^{14}$	$5.915\times 10^{14}$	$8.671\times 10^{-15}$

**Table 2 nanomaterials-12-03492-t002:** For a given value of η, the frequency position rules of the left (right) transmission (reflection) peaks with the increment of *G*, where *p* (*p* = 1, 2, …), *Q*, and *R* in the table are positive integers.

No	*η*	Coincident Peaks for Left and Right Incidence	Noncoincident Peaks in Same Period for Left Incidence	Noncoincident Peaks in Different Period for Left Incidence	Noncoincident Peaks in Same Period for Right Incidence	Noncoincident Peaks in Different Period for Right Incidence
1	1	$\nu_{G=3p}$	ν_G=3p__−__2_ < ν_G=3p__−__1_ < ν_G=3p_	ν_G=3p__−R_ < ν_G=3Q__−R_ (*Q* > *p*; *R* = 1, 2)	ν_G=3p__−__2_ > ν_G=3p-1_> ν_G=3p_	ν_G=3p__−R_ > ν_G=3Q__−R_ (*Q* < *p*; *R* = 1, 2)
2	2	$\nu_{G=2p}$	ν_G=2p__−__1_ < ν_G=2p_	ν_G=2p__−R_ < ν_G=2Q__−R_ (*Q* > *p*; *R* = 1)	ν_G=2p__−__1_ > ν_G=2p_	ν_G=2p__−R_ > ν_G=2Q__−R_ (*Q* < *p*; *R* = 1)
3	3	$\nu_{G=5p}$	ν_G=5p__−__4_ < ν_G=5p__−__3_ < ν_G=5p__−__2_ < ν_G=5p__−__1_ < ν_G=5p_	ν_G=5p__−R_ < ν_G=5Q__−R_ (*Q* > *p*; *R* = 1, 2, 3, 4)	ν_G=5p__−__4_ > ν_G=5p_ _−__3_ > ν_G=5p__−__2_ > ν_G=5p__−__1_ > ν_G=5p_	ν_G=5p__−R_ > ν_G=5Qv__R_ (*Q* < *p*; *R* = 1, 2, 3, 4)
4	4	$\nu_{G=3p}$	ν_G=3p__−__2_ < ν_G=3p__−__1_ < ν_G=3p_	ν_G=3p__−R_ < ν_G=3Q__−R_ (*Q* > *p*; *R* = 1, 2)	ν_G=3p__−__2_ > ν_G=3p-__−__1_ > ν_G=3p_	ν_G=3p__−R_ > ν_G=3Q__−R_ (*Q* < *p*; *R* = 1, 2)
5	5	$\nu_{G=5p}$	ν_G=5p__−__4_ < ν_G=5p__−__3_ < ν_G=5p__−__2_ < ν_G=5p__−__1_ < ν_G=5p_	ν_G=5p__−R_ < ν_G=5Q__−R_ (*Q* > *p*; *R* = 1, 2, 3, 4)	ν_G=5p__−__4_ > ν_G=5p_ _−__3_ > ν_G=5p__−__2_ > ν_G=5p__−__1_ > ν_G=5p_	ν_G=5p__−R_ > ν_G=5Qv__R_ (*Q* < *p*; *R* = 1, 2, 3, 4)
6	6	$\nu_{G=2p}$	ν_G=2p__−__1_ < ν_G=2p_	ν_G=2p__−R_ < ν_G=2Q__−R_ (*Q* > *p*; *R* = 1)	ν_G=2p__−__1_ > ν_G=2p_	ν_G=2p__−R_ > ν_G=2Q__−R_ (*Q* < *p*; *R* = 1)
7	7	$\nu_{G=3p}$	ν_G=3p__−__2_ < ν_G=3p__−__1_ < ν_G=3p_	ν_G=3p__−R_ < ν_G=3Q__−R_ (*Q* > *p*; *R* = 1, 2)	ν_G=3p__−__2_ > ν_G=3p-__−__1_ > ν_G=3p_	ν_G=3p__−R_ > ν_G=3Q__−R_ (*Q* < *p*; *R* = 1, 2)
8	8	$\nu_{G=5p}$	ν_G=5p__−__4_ < ν_G=5p__−__3_ < ν_G=5p__−__2_ < ν_G=5p__−__1_ < ν_G=5p_	ν_G=5p__−R_ < ν_G=5Q__−R_ (*Q* > *p*; *R* = 1, 2, 3, 4)	ν_G=5p__−__4_ > ν_G=5p_ _−__3_ > ν_G=5p__−__2_ > ν_G=5p__−__1_ > ν_G=5p_	ν_G=5p__−R_ > ν_G=5Qv__R_ (*Q* < *p*; *R* = 1, 2, 3, 4)

## Data Availability

The authors confirm that the data supporting the findings of this study are available within the article. All data supporting the findings of this study are available from the corresponding author (X.L.) on request.

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
