# Peer review of "Extraordinary Characteristics of One-Dimensional PT-Symmetric Ring Optical Waveguide Networks Composed of Adjustable Length Ratio Waveguides"

_nanomaterials, 2022, doi:10.3390/nano12193492_

Round 1

Reviewer 1 Report

Following their previous work on one-dimensional parity-time-symmetric periodic ring optical waveguide network (1D PTSPROWN), this manuscript optimized the simulation program and the length ratio of different materials within each unit-cell. Significant improvements were achieved after the optimization. The following comments need to be addressed before being accepted: 

1.     The theory part has many overlaps with their prior’s published paper “Extraordinary characteristics for one-dimensional parity-time-symmetric periodic ring optical waveguide networks ”. For example, figure 2, equations 3-7. I suggest the author properly cite previous work and only include necessary parts in a concise format. 

2.     What makes the dispersion relation (eq. (8)) different from the dispersion relation (eq. (22)) in the afore mentioned paper (ref. [5] in this manuscript)?

3.     Many information are missing, e.g., the 3rd and 4th affiliation, line 111, line 112, line 153, line 239, line 268, line 312, line 350.  The  author needs to proofread the manuscript carefully. 

4.     What’s the definition of a “vacuum waveguide”?

5.     Is that necessary for the upper and lower arm length ratio G be a positive integer? How about positive rational numbers or irrational numbers? 

Reviewer 2 Report

In this paper the authors present an optimization of their previously described 1-D parity-time-symmetric periodic ring optical waveguide network. Given that the theory has already been published, the novelty in this paper is the calculated performance of the optimized structure.

My main criticism of this work is that while it makes extraordinary claims for the performance of these devices, this is all based on numerical or analytical modeling of perfect structures, where lengths, refractive indices and gain values can be set to arbitrary precision. In order to provide more interest and value to the reader, we need to see an evaluation somewhere of the anticipated effect of non-ideal components. Furthermore, the extremely sharp resonances and high levels of localisation predicted here imply very large circulating optical powers. These will certainly introduce nonlinear changes in both gain and loss which will perturb the performance of these devices. Given that the basic theory has already been published, to my mind the next step has to be either an experimental demonstration (which would allow us to compare predicted and real performance) or a rigorous evaluation of the impact of real-world defects.

In terms of specific changes, I would therefore request:

1. An evaluation of the impact of expected non-idealities on the performance of the device

2. At least a discussion of the likely impact of non-linear optical effects

3. I find Table 2 very hard to understand - perhaps there is a clearer way of presenting that data?

4. There are several acronyms: APM, GPM, SPM which are not defined in the paper

Round 2

Reviewer 1 Report

Thank you for answering my questions. I still have several concerns with the revised version:

1. If Eq. (22) in the published ref.[5] is wrong, have you contacted the editor of Photonics Research and published a correction?

2. If G is not limited to be a positive integer, I hope the author include the explanations in the revised manuscript, instead of just answering the reviewers. Remember the final goal is to present results clearly to the global audience.

3. Most importantly, the main concern is not answered in the revised version. Experimental results or evaluation of non-ideal components should be added.

With these in mind, I think current version still needs imporvement and I will leave the final judge to the editor. 

Reviewer 2 Report

I appreciate that the authors have made their paper more readable. However, I still have significant concerns that I feel are not addressed. In particular, I do not see any acknowledgement in the revised paper that the very impressive results will be subject to non-idealities in the real world, such as imprecision in refractive index values and variation in the lengths of each section. Furthermore, although the authors state in their response to my previous review that nonlinear effects will be negligible since optical powers are low, in Section 3.1.2 they state that 'photonic localization' will approach a factor of 10^18. Unless I misunderstand something, this implies to me that very high local intensities would be generated. Similar limitations apply to the real-world performance of ultra-high Q-factor optical resonators, where non-linear effects need to be very carefully considered. As a result, I am not convinced that this paper has very much significance since I doubt that these results could be achieved in practice. A more thorough analysis of the sensitivity of the system to these non-idealities would be needed to help the reader understand how precisely everything would need to be engineered.
